# The Effect of the Treatment of Severe Early Childhood Caries on Growth-Development and Quality of Life

**DOI:** 10.3390/children10020411

**Published:** 2023-02-20

**Authors:** Ecenur Eyisoy Bagis, Sera Simsek Derelioglu, Fatih Sengül, Sinan Yılmaz

**Affiliations:** 1Department of Pedodontics, Faculty of Dentistry, Atatürk University, 25240 Erzurum, Turkey; 2Department of Public Health, Faculty of Medicine, Atatürk University, 25240 Erzurum, Turkey

**Keywords:** early childhood caries, ECOHIS, general anesthesia, growth and development, quality of life, oral health-related quality of life

## Abstract

Purpose: Untreated early childhood caries (ECC) adversely affect children’s quality of life. Our aim was to evaluate the effects of ECC on growth, development, and quality of life. Design and Methods: A total of 95 children were divided into three groups: general anesthesia (GA) (*n* = 31), dental clinic (DC) (*n* = 31), and control (*n* = 33). ECOHIS was applied to the parents in the GA and DC groups during a pre-treatment period and also applied in the post-treatment in the first and sixth months. Height, weight, and BMI measurements of the children in the study groups were taken and recorded at the pre-treatment stage and in the post-treatment in the first and sixth months. However, for the control group, these measurements were made just at the baseline and in the sixth month. Results: Upon the treatment of ECC, the total ECOHIS score significantly decreased (*p* < 0.001) in both groups in the following first month, whereas the scores of the children in the GA group reached a similar level to the DC group at the end of the sixth month. Following treatment, the weight and height of the children with ECC who initially had significantly lower BMI percentiles than the control group (*p* = 0.008) were observed to increase and, thus, they reached a similar BMI percentile value to the control group in the sixth month. Conclusions: The results of our study revealed that development and growth deficiencies in the children with ECC could be reversed rapidly by dental treatments and, thus, their quality of life would increase. The importance of treating ECC was revealed since treating ECC had positive effects both on the children’s growth and development and on the quality of life of the children and their parents.

## 1. Introduction

Early childhood caries (ECC) is a biofilm-mediated, multifactorial, dynamic disease induced by the imbalance between the demineralization and remineralization of dental hard tissues [1]. The American Academy of Pediatric Dentistry (AAPD) defines the ECC as the presence of a missing or filled tooth surface on any primary teeth of the children <6 years-old due to one or more cavitated/non-cavitated caries. [2] Although ECC was previously named “bottle tooth decay” since it was associated with bottle feeding, the term ECC has become popular over time by suggesting that the etiology of these types of caries was multifactorial. [2] The etiology of ECC consists of microbial factors (vertical transmission from the mother to the child), dietary habits (prolonged bedtime feeding), individual factors (salivary characteristics, hypoplasia), oral biofilm (oral hygiene habits and fluoride use), and socioeconomic factors [2,3,4,5,6]. ECC, which is regarded as a precursor of future caries development, remains a major heath issue with serious consequences despite the current preventive applications performed for avoiding the dental caries [7].

Oral health is complementary to general health and is a determinant factor for the quality of life. If ECC is not treated, it may cause many negative conditions, such as infection, pain, premature tooth loss, malnutrition, decrease in weight gain, sleeping problems, speech disorders, psychological and socioeconomic problems, missed school, and a decrease in the quality of life [8,9,10,11,12]. Oral health-related quality of life (OHRQoL) is a multi-dimensional concept that shows the individuals’ comfort in eating, sleeping, and communicating with the social circle and their self-confidence and satisfaction with their oral health [13]. The impact of oral diseases and treatments on the quality of life of children and their parents can be assessed using various scales. The early childhood oral health impact scale (ECOHIS) was developed to use on children aged between three and five. It was adapted into Turkish and was regarded as valid and reliable in Türkiye, and it has been concluded that it was a useful tool for the assessment of OHRQoL in children under the age of six [14,15].

Growth and development are significant indicators for children’s health, and although the evidence is controversial, it has been reported that dental caries affected anthropometric results related to growth. Infection due to untreated dental caries in children may cause pain and discomfort. Food intake may decrease when eating is painful [16,17]. It has been revealed that the mean weights of children with ECC were lower than the children without dental caries and the calorie intake of these children affected with ECC was insufficient [8,18,19]. Infection likely results in a decrease in appetite and food ingestion of the children and causes an inflammation-related energy consumption in the body, which all lead to growth retardation and subsequently failure to thrive (FTT). However, this situation is expected to reverse after a balanced diet [20]. Additionally, severe dental caries may cause restlessness and sleeping problems, which in turn may affect the quality of life. Inflammation caused by pulpitis and chronic abscesses may affect the growth through the same metabolic manner in which the cytokines affect erythropoiesis. A previous study reported non-significant increases in the mean weight of the children after the treatment of ECC under GA and remarked a significant improvement in the children’s quality of life as stated by their parents [16].

Based on this information, this study was conducted to examine the effect of the dental treatments of ECC-impacted children on their growth and development through the changes in their quality of life, height, body weight, body mass index (BMI), and percentile values and also to compare the collected data with those of the caries-free children. Two hypotheses were tested in the present study. 

 **Hypothesis 1.**
*Dental treatments performed did not affect the quality of life of the children with ECC.*


 **Hypothesis 2.**
*Percentile values of weight, height, and BMI of the ECC-affected children did not change after they were treated under GA or in the normal clinical conditions.*


## 2. Materials and Methods

The study included children with and without ECC who were admitted to the Atatürk University Faculty of Dentistry, Pediatric Dentistry Clinic.

This study is an interventional study including treatment and control groups. Study data were collected in a kindergarten and in the Atatürk University Faculty of Dentistry, Department of Pedodontics, in Erzurum between February 2019 and March 2020. The group treated under GA included the non-cooperative children with ECC. Children with systemic diseases and disabilities were excluded from all groups.

The independent variables in the study were determined as the sociodemographic characteristics related to social anamneses such as gender, age, environment, educational and income status of the parents; height, weight, BMI, and percentile values which were related to medical anamnesis; and occlusal relationship, dental abnormalities, halitosis, history of dental traumas, and bad oral habits which were related with dental anamnesis. On the other hand, dependent variables included ECOHIS scores; the presence of ECC among the children; and the indices of decayed, extracted, and filled teeth (deft), decayed, extracted, and filled tooth surface (defs), and oral health index-simplified (OHI-S).

The present study consisted of children with ECC but with no systemic diseases or disabilities. Uncooperative children who did not allow dental treatment in the clinical environment were included in the GA group while the cooperative ones were contained in the DC group. Additionally, the control group consisted of 4–6 years-old caries-free children.

The study originally consisted of 114 children aged between four and six years, and the control group had 42 children while the GA group had 33 and the DC group had 39. Because of failure to attend the follow-up visits at the first and sixth months and inability to reach the patients, 19 children were excluded from the study and only 95 were assessed. Thus, finally, the GA and DC groups had 31 children while the control group had 33. The flowchart of the study is given in Figure 1.

Parents were verbally informed about the study and asked to sign the informed consent forms. Children’s anthropometric measurements such as oral examinations and height, weight, and BMI measurements were performed. Weights were measured with a digital scale by observing the patients wearing light clothes without shoes. Moreover, children’s heights were measured by noticing that their heels, hips, shoulders, and heads formed a straight line against the wall and their heads should be positioned in the eye–ear plane (EEP). Children’s heights were marked on a vertical plane using a standard solid metric scale ruler. BMI values were calculated with the weight (kg)/[height (m)]^2^ formula [21]. The groups’ percentile categories were determined based on the reference values used in the anthropometric assessments for Turkish children promulgated in 2015 [22].

ECOHIS was administered to the parents in the study groups before the treatment. ECOHIS consists of a child impact section with four domains (symptoms, function, psychological, self-image/social interaction) including nine items and a family impact section with two domains (parent distress, family function) including four items. The answers given to the items aimed to reveal the frequency of children’s experience are as follows: 0 = never, 1 = rarely, 2 = occasionally, 3 = often, 4 = very often, 5 = I do not know. The total score is calculated by using the answers, and the maximum score is 52. Higher scores indicate more negative impacts on the quality of life. In the present study, the answer “I do not know” was regarded as the “missing answer”, and the scales with missing answers ≥2 and ≥1 in the children and family impact sections, respectively, were excluded [14,15]. Anthropometric parameters were remeasured with ECOHIS in the follow-up visits performed at the pre and post-treatment first and sixth months; these measurements were conducted in the control group just at the baseline and in the sixth month.

The fact that all the participating children were from the city of Erzurum might imply that the results covered only the study area, constituting the greatest limitation of this study. The scope of the present study can be expanded with different future studies to be conducted in the different regions by increasing the sample size.

All data were entered into the Statistical Package for Social Sciences (SPSS) version 20. Numerical values’ goodness of fit was examined using the Kolmogorov–Smirnov Test, and z values calculated for skewness and kurtosis were also computed with graphical methods. Mean values of the age and height following normal distribution were compared using the one-way analysis of variance (one-way ANOVA) and post hoc Duncan multiple comparison tests in the independent groups, and the Paired Samples *t*-Test was used in the dependent groups. Kruskal–Wallis and Mann–Whitney U tests were used for the inter-group comparisons of numerical variables following a non-normal distribution. Moreover, Friedman’s two-way ANOVA test developed for the dependent groups was used to determine the differences between the values measured at different stages of the study.

In all ECOHIS tables, different letters in the same lines and columns were statistically different. Lines indicated the intra-group (a, b, c) differences while columns reflected the intergroup differences (x, y, z). The level of significance for the analytic results was set as *p* < 0.05.

## 3. Results

The total number of children assessed in the study was 95. Both GA and DC groups included 31 children while the control group consisted of 33 (Table 1). The groups had no difference in terms of the distribution of gender (*p* > 0.05). A statistically significant difference regarding the mean deft and defs values was found between the GA and DC groups (*p* = 0.014, *p* < 0.001) (Table 2). The mean OHI-S values in GA and DC groups were similar whereas the control group’s mean value of OHI-S was statistically and significantly lower than these groups (*p* < 0.001).

The number of parents with a bachelor’s degree or higher was elevated in the control group. An assessment of the income status indicated that 52.6% of children were from moderate-income families and the parents of the children in the control group had higher levels of income.

None of the parents to whom ECOHIS was administered selected the answer of “I do not know”. In all groups, “pain, sensitivity to hot or cold beverages, difficulty in eating, problems in sleeping, and being irritable or frustrated” were the most responded to items in the children impact section. Additionally, “being upset and feeling guilty” were the items most responded to in the family impact section in all groups. Mean scores of ECOHIS items in GA and DC groups obtained at three stages of the study are presented in Table 3.

The pre-treatment, first-, and sixth-month ECOHIS scores of the study groups are presented in Table 4. An assessment of the ECOHIS scores obtained on the child impact section indicated that scores in the child symptoms domain in the pre-treatment stage significantly decreased in both groups in the first month (*p* < 0.001) with a greater decline in the GA group than DC (*p* = 0.049), whereas the scores of two groups were equalized in the sixth month (*p* = 1). With regard to the pre-treatment scores, in the domain of child function, a greater decrease was observed in the DC group than the GA (*p* < 0.001) in both post-treatment first- and sixth-month follow-ups (*p*_1 month_ = 0.033, *p*_6 month_ = 0.025). In the child psychology domain, there was a greater decline in the ECOHIS scores of the DC group than the GA in the first month regarding the baseline scores (*p* < 0.001). In the child self-confidence domain, a significant decrease was observed only in the GA group in the sixth month, compared to the scores obtained at the pretreatment stage (*p* < 0.001) where a significant difference was also detected between the scores of DC and GA groups just in the first-month follow-up (*p* = 0.011). Baseline scores in the child score domain decreased significantly in both groups in the first month (*p* < 0.001).

Evaluation of the ECOHIS scores obtained on the family impact section revealed that in the parental distress domain, baseline scores decreased significantly in the GA group in the sixth month, whereas they decreased in the DC group in the first month (*p* < 0.001). In the first month, pre-treatment scores in the family function domain decreased significantly in both groups (*p* < 0.01) with a higher decline in the DC group (*p*_1mont_ = 0.002). In the family score domain, initial scores decreased significantly in GA and DC groups in the sixth and first months, respectively (*p* < 0.001), where there was also a significant difference between the DC and GA groups due to the decrease in the DC group in the first month (*p* = 0.008).

When the total ECOHIS score was assessed, regarding the pre-treatment scores, a significant decrease was observed in both groups in the first month (*p* < 0.001) with a higher decline in the DC group (*p*_1 month_ = 0.002), whereas the scores of the GA group decreased significantly (*p* < 0.001) and reached a similar value to the DC group in the sixth month (*p*_6 month_ = 0.681).

The groups’ anthropometric and percentile values measured at the different stages of the study are given in Table 5. Gains in weight and height were observed over time in the groups consisting of children who were in the growth and development stage (*p* < 0.001). Moreover, in regard to the weight-for-height percentiles of the groups, there were statistically significant increases between the pre-treatment and sixth month follow-up values (*p* < 0.001). However, no significant difference was observed between the group scores measured at the different study stages. BMI scores did not differ over time in the control group (*p* = 0.673), whereas they increased in the GA and DC groups in all stages of measurement (*p* < 0.001). As for the BMI percentiles, although the GA and DC groups had lower baseline percentile values than the control group (*p* = 0.008), they reached similar percentiles (*p* = 0.288) to the control group due to the increases observed at the end of the sixth month (*p* < 0.001).

## 4. Discussion

The consequences of ECC have a great impact on general health. Untreated caries-associated infection may cause various issues such as pain, difficulties in eating, anxiety, trouble sleeping, or psychological problems. These issues affect quality of life but OHRQoL can be improved through dental treatments [1,23]. In addition, deficiencies in the intake of nutrients, which are necessary for growth, may arise when eating becomes painful due to caries [16,17]. These issues may result in malnutrition and growth disorders. A normal growth pattern is an expected result of improved nutrition following the dental treatment. Previous studies reported a relationship between untreated carries-associated problems in young children and insufficient growth [8,12,18].

ECC impacts more than 600 million children worldwide and mostly remains untreated [1]. Regarding child behavior management, treatment of ECC is as challenging today as it was previously. For an effective treatment, firstly, children, parents, and dentists should cooperate well. There are several methods and techniques regularly used in ECC treatment, such as behavior control management performed in routine dental conditions and advanced behavior guidance techniques (protective stabilization, conscious sedation, deep sedation, and general anesthesia). [24,25] Most children with ECC can be effectively treated with basic behavior management techniques. However, treatment of younger children with ECC may not be always peaceful in normal clinical conditions. Moreover, children with disorders of psychological development and non-cooperative children due to some physical, mental, and medical challenges may need to be treated using advanced techniques. Comprehensive rehabilitation under GA is the only option to provide high-quality dental treatment for non-cooperative children with severe ECC [26]. In the present study, dental treatments of the non-cooperative children (GA group) were performed under GA, whereas the cooperative ones (DC group) were treated in routine clinic conditions.

The consequences of untreated ECC have a negative impact on children’s quality of life and on their oral and dental health and eventually may pose a heavy socioeconomic burden to their parents [27]. Previous studies reported that dental treatments of infants and toddlers with multiple caries performed under GA had significant effects on both their and their families’ quality of life [28,29]. Poor oral health in children is closely associated with growth retardation and developmental delay since oral health affects general health [30].

Similar to the present study, another research conducted to evaluate the impact of dental treatments on children’s health with different perspectives indicated that treating severe dental caries in the child population significantly decreased pain, lesions, dissatisfaction, smiling avoidance, and anorexia. Additionally, although the differences in the development of children were expected between the groups, they were not statistically significant [31]. Contrary to the present study, the aforementioned study did not provide any information about the impacts of dental treatments on children’s anthropometric values since no long-term follow-ups were involved. There is a limited number of studies in the literature examining weight gains among children with ECC after the dental treatments performed under GA and also assessing the changes in these children’s quality of life [16,32]. However, many studies had examined these two parameters separately. The current study was planned since there were no studies in the literature comparing the impact of severe ECC on the growth and quality of life with those of healthy caries-free children.

The current study investigated the effects of ECC treatment performed under GA and in routine clinic conditions on growth and development and quality of life. Thus, the findings of oral examinations of the children aged 4–6 years and their anthropometric findings were collected with ECOHIS. This questionnaire was reported to be repeated periodically in order to thoroughly determine the post-treatment effects and to evaluate whether the improvement in the quality of life was sustainable [33]. For this reason, scores of ECOHIS and anthropometric values were recorded with the measurements performed at the baseline, first, and sixth months. Children in the control group were excluded from ECOHIS since they did not have any caries. At the same time, anthropometric measurements of these children were performed only at the baseline and in the sixth month since the chances in their first month measurements would not be significantly different.

Based on the present ECOHIS results, the items most commonly responded to in the child impact section included pain and difficulty in eating. Difficulty in consuming hot or cold beverages, being irritable and frustrated, and trouble sleeping were among the other commonly responded items, which were also reported by the previous studies conducted using ECOHIS [23,26]. The least commonly responded item in the child impact section in both groups was talking avoidance. Additionally, smile or laughing avoidance and having difficulties pronouncing any words were the other less commonly responded items. The least affected and altered domain in the pre-treatment stage was child self-confidence in both groups. These results might have arisen from the fact that children aged between four and six years had less social consciousness and cognitive development, which made them less sensitive to the impacts of social factors such as talking or smiling. In the present study, a low baseline score and minimal post-treatment changes in the child self-confidence domain were found to be similar to those of the previous studies [32,34]. The most commonly responded items were observed in the parental distress domain of the family impact section in both groups. The literature review revealed some previous studies with similar results to the family impact section of the present study. Furthermore, the decreases observed in the post-treatment ECOHIS scores of the current study were found to be in parallel with those of other studies evaluating the impacts of dental treatments performed under GA in the follow-up stages [23,26,35].

Higher total ECOHIS score of the GA group than the DC might be attributed to the domain of parental distress, and primarily to the family score. Significant decreases were observed in the total ECOHIS scores of both groups at the end of the 1st month whereas GA group reached a similar score to DC in the 6th month.

A higher total ECOHIS score in the GA group than the DC group in the first month might have arisen from the longer recovery times due to more aggressive treatment methods, such as performing multiple procedures under GA, stainless steel crown (SSC) placement, and tooth extraction. Similar to the present study, a previous study reported that dental treatments performed under GA had significantly improved both the children’s and their families’ quality of life and their oral and dental health, as well as had some positive impacts on their physical and psychosocial health and welfare. Hypothesis 1 set as “dental treatments performed did not affect the quality of life of the children with ECC” was rejected due to the lower ECOHIS scores observed in the follow-ups of the treated children.

A previous study assessing the BMIs of the ECC-affected children treated under GA reported that the majority of them were underweight [30]. Acs et al. remarked that the weight percentile values of the children with ECC had reached the same values as the children who became caries-free after the comprehensive dental treatments performed under GA [18]. Improving the OHRQoL of the child patients with higher baseline ECOHIS scores by recovering their poor digestive functions with the dental treatments was believed to be effective in the significant increase in the weight, height, and BMI percentiles of the groups observed at the post-treatment sixth month. Hypothesis 2 set as “percentile values of weight, height and BMI of the ECC-affected children did not change after they were treated under GA or in the clinical conditions” was also rejected since anthropometric scores of the children treated under GA and in routine clinical conditions reached the same levels as the healthy ones.

A previous study evaluating weight changes and weight-related percentile values of the children treated under GA revealed that the mean weight of the children with dental caries was not under the 50th percentile and also reported that the slight increases in weight following the dental treatment were not an indicator of growth, contrary to what was found in the present study and in many other relevant studies. However, based on the parental report, that study indicated that children’s quality of life improved significantly [16]. Another study examining the impact of dental treatments with extraction used for the carious teeth with pulp involvement on the weight and height of the children indicated a significant weight gain following the treatment [30]. Thus, when planning actions to improve growth, untreated dental caries should always be taken into consideration as an important factor affecting the growth of children.

According to the results of the present study, ECC had an adverse impact on the quality of life related to dental health and on growth development. Moreover, these two conditions were closely related; the better the OHRQoL of children and parents, the more positive and the better the development and growth of the children in this highly wealthy environment.

ECC leads to some esthetical, phonetic, functional, and psychological problems in early childhood and adversely affects growth and development. In the present study, by researching the impact of ECC on child growth and development, it was revealed that following the treatment of ECC, the growth and development gap between the ECC-affected children and healthy ones might close rapidly and the children’s quality of life might be improved. Results of this study also showed that a stronger public effort was needed for preventing ECC; thus, ECC prevention should absolutely be incorporated into public health training programs. Consequently, it was concluded that not only the efforts of pediatric dentists but also the holistic efforts of all health care providers were required.

## 5. Conclusions

In our study, OHRQoL had significantly increased since the first month due to the dental treatments of the ECC-affected children. Furthermore, these children reached the standards of the healthy children of the same age as a result of the increases in their weight and BMI percentiles at the end of the sixth month. The study data clearly reflected the effects of providing the best possible oral and dental health for the children on their growth and development, emphasized the correlation between general and oral health, and highlighted the importance of perceiving the impacts of ECC on both the children and their families’ quality of life. Moreover, it showed the relationship between the growth and development and a multi-factor concept such as quality of life. Accordingly, the significance of ECC should be well understood, and the highest priority should be given to the implementation of preventative measures and parental and social awareness should also be raised.

## Figures and Tables

**Figure 1 children-10-00411-f001:**
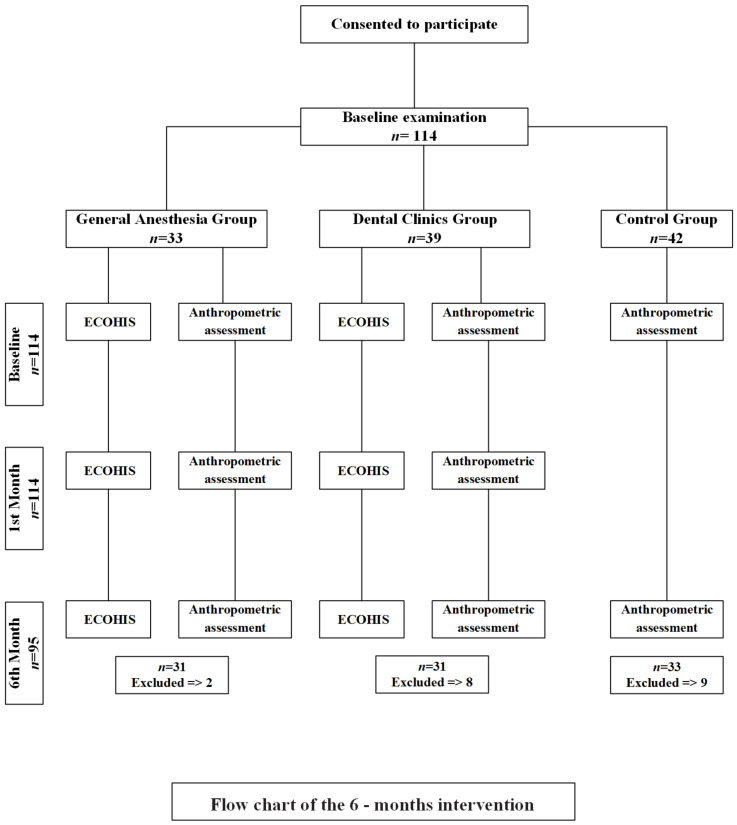
Flowchart of the study.

**Table 1 children-10-00411-t001:** Distribution of gender by the groups.

	Girl (*n*, %)	Boy (*n*, %)	Total (*n*, %)
General Anesthesia (GA)	17 (54.8)	14 (45.2)	31 (32.6)
Dental Clinic (DC)	17 (54.8)	14 (45.2)	31 (32.6)
Control (C)	16 (48.5)	17 (51.5)	33 (34.7)
Total	50 (52.6)	45 (47.3)	95 (100)

**Table 2 children-10-00411-t002:** Mean deft, defs, and OHI-S indices of the groups (Mean ± Standard Deviation).

	General Anesthesia	Dental Clinic	Control	*p*
deft	12.48 ± 2.62	10.48 ± 3.46	0	0.014
defs	30.42 ± 9.99	19.29 ± 7.84	0	<0.001
OHI-S	1.71 ^b^ ± 0.47	1.35 ^b^ ± 0.41	0.52 ^a^ ± 0.31	<0.001

^a,b^: The means marked with different letters in the same row are statistically different (*p* < 0.05). OHI-S: oral health index-simplified

**Table 3 children-10-00411-t003:** Mean scores of ECOHIS items in GA and DC groups obtained at the different stages (Mean ± Standard Deviation).

	General Anesthesia			Dental Clinic		
	Pretreatment	1st Month	6th Month	Pretreatment	1st Month	6th Month
(1) How often did your children have pain in their teeth, mouths, or chins?	2.9 ^a^ ± 0.9	0.3 ^b^ ± 0.6	0.3 ^b^ ± 0.5	3.1 ^a^ ± 1.2	0.6 ^b^ ± 0.7	0.3 ^b^ ± 0.1
(2) How often did your children have difficulties consuming hot or cold beverages due to their dental problems and treatments?	2.6 ^a^ ± 1.2	0.2 ^b^ ± 0.4	0.1 ^b^ ± 0.3	2.7 ^a^ ± 1.2	0.1 ^b^ ± 0.4	0 ^b^ ± 0
(3) How often did your children have difficulties consuming certain foods due to their dental problems and treatments?	2.8 ^a^ ± 1.2	0.4 ^b^ ± 0.8	0.2 ^b^ ± 0.4	2.9 ^a^ ± 1.2	0.2 ^b^ ± 0.4	0.1 ^b^ ± 0.2
(4) How often did your children have difficulties pronouncing a word due to their dental problems and treatments?	0.8 ^a^ ± 1.4	0.1 ^b^ ± 0.4	0.1 ^b^ ± 0.3	0.4 ^a^ ± 0.9	0 ^b^ ± 0	0.1 ^b^ ± 0.4
(5) How often did your children miss kindergarten or nursery school due to their dental problems and treatments?	1 ^a^ ± 1.3	0.2 ^b^ ± 0.6	0 ^b^ ± 0	0.7 ^a^ ± 1	0.1 ^b^ ± 0.4	0 ^b^ ± 0
(6) How often did your children have trouble sleeping due to their dental problems and treatments?	2.4 ^a^ ± 1.2	0.2 ^b^ ± 0.4	0.1 ^b^ ± 0.2	1.8 ^a^ ± 1.5	0 ^b^ ± 0	0.1 ^b^ ± 0.4
(7) How often did your children become irritable or frustrated due to their dental problems and treatments?	2.7 ^a^ ± 1.1	0.5 ^b^ ± 0.7	0.1 ^c^ ± 0.1	2.2 ^a^ ± 1.6	0 ^b^ ± 0	0.2 ^b^ ± 0.5
(8) How often did your children avoid smiling or laughing due to their dental problems and treatments?	0.8 ^a^ ± 1.2	0.2 ^b^ ± 0.7	0.1 ^b^ ± 0.4	0.4 ^a^ ± 0.9	0 ^b^ ± 0	0 ^b^ ± 0.2
(9) How often did your children avoid talking due to their dental problems and treatments?	0.5 ^a^ ± 1	0.1 ^b^ ± 0.3	0 ^b^ ± 0	0.3 ^a^ ± 0.9	0 ^b^ ± 0	0 ^b^ ± 0.2
(10) How often did you or other family members become upset by the dental problems and treatments of your children?	2.9 ^a^ ± 1.3	1.5 ^b^ ± 1.4	0.6 ^c^ ± 0.7	3.8 ^a^ ± 0.6	0.9 ^b^ ± 1.3	0.4 ^c^ ± 0.6
(11) How often did you or other family members feel guilty due to the dental problems and treatments of your children?	3 ^a^ ± 1.4	2.7 ^a^ ± 1.3	2 ^b^ ± 1.3	3.6 ^a^ ± 0.9	2.5 ^b^ ± 1.2	2.3 ^b^ ± 1.2
(12) How often did you or other family members take a day off from work due to the dental problems and treatments of your children?	2.1 ^a^ ± 1.3	1.1 ^b^ ± 1.1	0.6 ^b^ ± 0.7	1.1 ^a^ ± 1.5	0.1 ^b^ ± 0.4	0.5 ^b^ ± 0.8
(13) How often did you or other family members have financial problems due to the dental problems and treatments of your children?	0.6 ^a^ ± 1.1	0.5 ^a^ ± 1	0.4 ^a^ ± 0.7	0.5 ^a^ ± 0.9	0.4 ^a^ ± 1	0.4 ^a^ ± 0.8

^a,b,c^: The means marked with different letters in the same row are statistically different (*p* < 0.05).

**Table 4 children-10-00411-t004:** Distribution of ECOHIS scores of the study groups in the first and sixth months (Mean ± Standard Deviation).

		Pretreatment	First Month	Sixth Month	*p*
Child Impact Scale					
Child symptoms	GA	2.9 ^a^ ± 0.9	0.3 ^b,x^ ± 0.6	0.3 ^b^ ± 0.5	<0.001
	DC	3.1 ^a^ ± 1.2	0.6 ^b,y^ ± 0.7	0.3 ^b^ ± 0.5	<0.001
	*p*	0.281	0.049	1	
Child function	GA	7.2 ^a^ ± 3	0.9 ^b,y^ ± 1.2	0.4 ^b,y^ ± 0.6	<0.001
	DC	6.7 ^a^ ± 3	0.4 ^b,x^ ± 0.8	0.1 ^b,x^ ± 0.4	<0.001
	*p*	0.777	0.033	0.025	
Child psychology	GA	5 ^a^ ± 2	0.7 ^b,y^ ± 0.9	0.1 ^b^ ± 0.3	<0.001
	DC	4 ^a^ ± 2.9	0 ^b,x^ ± 0	0.2 ^b^ ± 0.8	<0.001
	*p*	0.094	<0.001	0.663	
Child self-confidence	GA	1.3 ^a^ ± 2	0.3 ^a,b,y^ ± 0.7	0.1 ^b^ ± 0.4	<0.001
	DC	0.7 ± 1.6	0 ^x^ ± 0	0.1 ± 0.3	1
	*p*	0.086	0.011	0.583	
Child Score	GA	16.5 ^a^ ± 5.8	2.1 ^b^ ± 2.4	0.8 ^b^ ± 1.1	<0.001
	DC	14.5 ^a^ ± 7.3	1 ^b^ ± 1.3	0.7 ^b^ ± 1.1	<0.001
	*p*	0.842	0.084	0.625	
Family Impact Scale					
Parental Distress	GA	5.9 ^a,x^ ± 2.6	4.2 ^a^ ± 2.4	2.6 ^b^ ± 1.6	<0.001
	DC	7.4 ^a,y^ ± 1.4	3.4 ^b^ ± 2.1	2.7 ^b^ ± 1.4	<0.001
	*p*	0.003	0.099	0.686	
Family Function	GA	2.7 ^a,y^ ± 1.9	1.6 ^b,y^ ± 1.7	0.9 ^b^ ± 1.1	<0.001
	DC	1.6 ^a,x^ ± 1.7	0.5 ^b,x^ ± 1.1	0.8 ^ab^ ± 1.5	0.009
	*p*	0.024	0.002	0.193	
Family Score	GA	8.6 ^a^ ± 3.9	5.7 ^a,y^ ± 2.8	3.5 ^b^ ± 2.1	<0.001
	DC	9 ^a^ ± 2.3	3.8 ^b,x^ ± 2.6	3.6 ^b^ ± 2.3	<0.001
	*p*	0.645	0.008	0.887	
Total Score	GA	25 ^a^ ± 8	7.8 ^b,y^ ± 3.7	4.3 ^c^ ± 2.5	<0.001
	DC	23.5 ^a^ ± 8.5	4.8 ^b,x^ ± 2.9	4.2 ^b^ ± 2.7	<0.001
	*p*	0.571	0.002	0.681	

^a,b,c^: Different letter-marked averages in the same row are statistically different (*p* < 0.001). ^x,y^: The means marked with different letters in the same column are statistically different (*p* < 0.05). GA: General Anesthesia; DC: Dental Clinic; C: Control.

**Table 5 children-10-00411-t005:** Anthropometrical and percentile values of the study groups measured at the different stages of the study (Mean ± Standard Deviation).

		Pretreatment	First Month	Sixth Month	*p*
Height (cm)	GA	106.6 ^a,y^ ± 5.8	107.1 ^b,x^ ± 5.8	111 ^c^ ± 5.9	<0.001
	DC	110.1 ^a,y^ ± 5	110.8 ^b,y^ ± 4.5	114 ^c^ ± 4.9	<0.001
	C	105.5 ^a,x^ ± 7.6		111.4 ^b^ ± 6.8	<0.001
	*p*	0.012	0.008	0.098	
Height Percentile (%)	GA	38 ^a^ ± 30	42.4 ^a^ ± 30	54.7 ^b^ ± 30.4	<0.001
	DC	38.8 ^a^ ± 31.8	44.2 ^ab^ ± 32	51.2 ^b^ ± 29.9	<0.001
	C	36.2 ^a^ ± 32.9		54.6 ^b^ ± 29.9	<0.001
	*p*	0.947	0.822	0.874	
Weight (kg)	GA	17.7 ^a^ ± 2.6	18.3 ^b^ ± 2.7	20.1 ^c^ ± 3	<0.001
	DC	18.5 ^a^ ± 2.4	19 ^b^ ± 2.5	20.7 ^c^ ± 2.8	<0.001
	C	18.1 ^a^ ± 2.9		20.2 ^b^ ± 3	<0.001
	*p*	0.459	0.295	0.735	
Weight Percentile (%)	GA	40 ^a^ ± 28.9	46.2 ^a^ ± 28.7	57.4 ^b^ ± 26.7	<0.001
	DC	40.4 ^a^ ± 29.4	44.2 ^a^ ± 29	54.8 ^b^ ± 28.7	<0.001
	C	47.1 ^a^ ± 29.7		58 ^b^ ± 25.6	<0.001
	*p*	0.552	0.782	0.884	
BMI (kg/m^2^)	GA	15.5 ^a,x,y^ ± 1.7	15.9 ^b^ ± 1.8	16.3 ^b^ ± 1.9	<0.001
	DC	15.3 ^a,x^ ± 1.8	15.5 ^b^ ± 1.8	15.9 ^c^ ± 1.9	<0.001
	C	16.3 ^y^ ± 1.7		16.2 ± 1.4	0.673
	*p*	0.044	0.302	0.603	
BMI Percentile (%)	GA	44.5 ^a,x^ ± 31.6	50.5 ^a^ ± 30.3	60.2 ^b^ ± 27.9	<0.001
	DC	40.8 ^a,x^ ± 30	46.8 ^ab^ ± 31	52.9 ^b^ ± 29.8	<0.001
	C	62.9 ^y^ ± 28		63.6 ± 24.6	0.718
	*p*	0.008	0.636	0.288	

^a,b,c^: Different letter-marked averages in the same row are statistically different (*p* < 0.001). ^x,y^: The means marked with different letters in the same column are statistically different (*p* < 0.05). GA: General Anesthesia; DC: Dental Clinic; C: Control.

## Data Availability

The data that support the findings of this study are available from the corresponding author upon reasonable request.

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
