# Peer review of "The Effect of the Treatment of Severe Early Childhood Caries on Growth-Development and Quality of Life"

_children, 2023, doi:10.3390/children10020411_

Round 1

Reviewer 1 Report

The aim of these study as presented in lines 41-44, is to compare the Quality of life between Severe ECC children and caries free, pre and post treatment.  The study design however does not investigate this hypothesis. in some sections it includes comparison of GA with DC treated children and elsewhere with caries free.

Tables are confusing, incomplete non matching titles, and  content of columns.  some of the descriptive data are of no interest, canbe added as Appendix.

In many sections the term "domestic impact section" is used, but it is not  described in the M and Materials. 

Very poorly written paper of an interesting subject.  Needs restructuring of the data, reanalysis and all sections except introduction to be rewritten.

Author Response

The aim of these study as presented in lines 41-44, is to compare the Quality of life between Severe ECC children and caries free, pre and post treatment.  The study design however does not investigate this hypothesis. in some sections it includes comparison of GA with DC treated children and elsewhere with caries free.

Hypotheses were added.

Tables are confusing, incomplete non matching titles, and  content of columns.  some of the descriptive data are of no interest, canbe added as Appendix.

Title of Table 2 were amended as “Mean dft, dfs and OHI-S indices of the groups (Mean ± Standard Deviation)”. Table 5 was removed.

In many sections the term "domestic impact section" is used, but it is not  described in the M and Materials. 

Corrected as “Family impact section”.

Very poorly written paper of an interesting subject.  Needs restructuring of the data, reanalysis and all sections except introduction to be rewritten.

Article was amended and rewritten in accordance with the  recommendations and corrections of the reviewers.

Reviewer 2 Report

The MS is interesting, however the he presentation must be improved. Please describe the groups in detail, present the assessed dmf/t and dmf/s, and characterize the treatment procedures and the non-treated (?) control group.

Present in "Results" the contents and not only the scores, statistically different scores make no sense, different outcomes make sense.

Please use clear scientific formulations in "Conclusions": What is the main outcome? What concept was revealed?

Please  correct the "References" strictly according to the author`s instructions.

Author Response

The MS is interesting, however the he presentation must be improved. Please describe the groups in detail, present the assessed dmf/t and dmf/s, and characterize the treatment procedures and the non-treated (?) control group.

dmf/t and dmf/s finding are given in Table 2.

The present study consisted of the healthy children with ECC, with no systemic diseases and disabilities. Uncooperative children who did not allow dental treatment in the clinical environment were included in the GA group while the cooperative ones were contained in the DC group. And the control group consisted of 4-6 years-old caries-free children.

Present in "Results" the contents and not only the scores, statistically different scores make no sense, different outcomes make sense.

Results section was rewritten.

Please use clear scientific formulations in "Conclusions": What is the main outcome? What concept was revealed?

Corrected

Please  correct the "References" strictly according to the author`s instructions.

Corrected

Round 2

Reviewer 1 Report

Table 1 and 2 still need the titles revised

Author Response

Table titles were corrected.

The article has been revised again in terms of grammar.

Reviewer 2 Report

Thank you for your corrections, the MS has improved now. However, please correct Table 2: dft means dmf/t and dmf/s? OHI-S is ECOHIS? Please present in Tables 4 and 5 ALL abbreviations in the legends! The text needs some minor corrections, please go carefully through the whole text!

Author Response

  • Tables were corrected. (dft was corrected as deft.)
  • OHI-S is not ECOHIS. OHI-S is oral health index - simplified.
  • Tables 4 and 5 were corrected. due to insufficient table cell spaces general anesthesia, dental clinic, control were abbreviated as GA, DC, C and they were explained as footnotes.
  • It was reviewed and corrected.
